# Extracting Strong Policies for Robotics Tasks from Zero-Order Trajectory Optimizers

**Cristina Pinneri**[1,2][*] **Shambhuraj Sawant**[1,3][*]**, Sebastian Blaes**[1]**, and Georg Martius**[1]

[1]Autonomous Learning Group, Max Planck Institute for Intelligent Systems, Tübingen, Germany
[2]Department of Computer Science, ETH Zurich and Max Planck ETH Center for Learning Systems
[3]Department of Engineering Cybernetics, NTNU, Trondheim, Norway
`{cpinneri,ssawant,sblaes,gmartius}@tuebingen.mpg.de`

## ABSTRACT

Solving high-dimensional, continuous robotic tasks is a challenging optimization problem. Model-based methods that rely on zero-order optimizers like the cross-entropy method (CEM) have so far shown strong performance and are considered state-of-the-art in the model-based reinforcement learning community. However, this success comes at the cost of high computational complexity, being therefore not suitable for real-time control. In this paper, we propose a technique to jointly optimize the trajectory and distill a policy, which is essential for fast execution in real robotic systems. Our method builds upon standard approaches, like guidance cost and dataset aggregation, and introduces a novel adaptive factor which prevents the optimizer from collapsing to the learner's behavior at the beginning of the training. The extracted policies reach unprecedented performance on challenging tasks like making a humanoid stand up and opening a door without reward shaping.

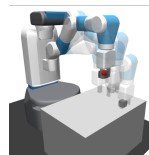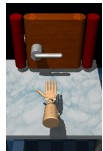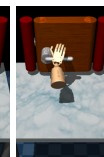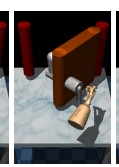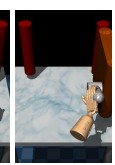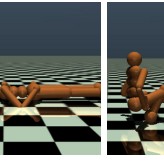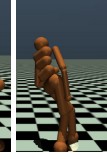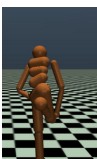

Figure 1: Environments and exemplary behaviors of the learned policy using APEX. From left to right: FETCH PICK&PLACE (sparse reward), DOOR (sparse reward), and HUMANOID STANDUP.

## 1 INTRODUCTION

The general purpose of model-based and model-free reinforcement learning (RL) is to optimize a trajectory or find a policy that is fast and accurate enough to be deployed on real robotic systems.

Policies optimized by model-free RL algorithms achieve outstanding results for many challenging domains (Heess et al., 2017; Andrychowicz et al., 2020), however, in order to converge to the final performance, they require a large number of interactions with the environment and can hardly be used on real robots, which have a limited lifespan. Moreover, real robotic systems are high-dimensional and have a highly non-convex optimization landscape, which makes policy gradient methods prone to converge to locally optimal solutions. In addition, model-free RL methods only gather task-specific information, which inherently limits their generalization performance to new situations.

On the other hand, recent advances in model-based RL show that it is possible to match model-free performance by learning uncertainty-aware system dynamics (Chua et al., 2018; Deisenroth & Rasmussen, 2011; Du et al., 2019). The learned model can then be used within a model-predictive control framework for trajectory optimization. Zero-order optimizers are gaining a lot of traction in

---

[*]equal contribution.
We acknowledge the support from the German Federal Ministry of Education and Research (BMBF) through the Tübingen AI Center (FKZ: 01IS18039B) and from the Max Planck ETH Center for Learning Systems.

the model-based RL community (Chua et al., 2018; Wang & Ba, 2020; Williams et al., 2015) since they can be used with any choice of model and cost function, and can be surprisingly effective in finding high-performance solutions (Pinneri et al., 2020) (close to a global optimum) in contrast to their gradient-based counterparts, which are often highly dependent on hyperparameter tuning (Henderson et al., 2017). One of the most popular optimizers is the Cross-Entropy Method (CEM), originally introduced in the 90s by Rubinstein & Davidson (1999).

Despite their achievements, using zero-order methods for generating action sequences is time consuming in complex high-dimensional environments, due to the extensive sampling, making it hard to deploy them for real-time applications.

Extracting a policy from powerful zero-order optimizers like CEM would bridge the gap between model-based RL in simulation and real-time robotics. As of today, this is still an open challenge (Wang & Ba, 2020).

We analyze this issue and showcase several approaches for policy extraction from CEM. In particular, we will use the sample-efficient modification of CEM (iCEM) presented in Pinneri et al. (2020). Throughout the paper, we will call these optimizers "experts" as they provide demonstration trajectories. To isolate the problem of bringing policy performance close to the expert's one, we consider the true simulation dynamics as our forward model.

Our contributions can be summarized as follows:

- pinpointing the issues that arise when trying to distill a policy from a multimodal, stochastic teacher;
- introducing APEX, an Adaptive Policy EXtraction procedure that integrates iCEM with DAgger and a novel adaptive variant of Guided Policy Search;
- our specific integration of methods produces an improving adaptive teacher, with higher performance than the original iCEM optimizer;
- obtaining strong policies for hard robotic tasks in simulation (HUMANOID STANDUP, FETCH PICK&PLACE, DOOR), where model-free policies would usually just converge to local optima.

Videos showing the performance of the extracted policies and other information can be found at `https://martius-lab.github.io/APEX`.

## 2 RELATED WORK

Our objective is to extract high-performing policies from CEM experts that can operate with a few planning samples to make iterative learning fast. Other kinds of zero-order optimizers have been used to generate control sequences (Williams et al., 2015; Lowrey et al., 2019) but they still have to evaluate thousands of trajectories for each time step. Even simple random shooting has been used as a trajectory optimizer to bootstrap a model-free policy (Nagabandi et al., 2018).

To train policies from optimal control solutions, it was shown that the expert optimizers need to be guided towards the learning policy – known as guided policy search (GPS) (Levine & Koltun, 2013; Levine & Abbeel, 2014). In our work, the expert does not come from optimal control but is the stochastic iCEM optimizer, which we will also refer to as *teacher*. We apply GPS in a model model-predictive control setting, as done in Levine & Koltun (2013); Mordatch & Todorov (2014); Mordatch et al. (2015); Zhang et al. (2016); Kahn et al. (2017); Sun et al. (2018) using local trajectory optimization to generate a dataset for training a global policy through imitation learning. These approaches alternate between training a policy and creating new data with a model-based supervisor guided towards the learner, which was formalized in Sun et al. (2018). Stochastic experts require particular guidance strategies, such as an **adaptive cost** formulation that we propose here, together with expert warm-starting via distribution initialization and additional samples from the policy. A simple form of warm-starting was already done in Wang & Ba (2020).

Recently, approaches like simple point-to-point supervised training such as Behavioral Cloning (BC), or Generative Adversarial Network training (GAN) have been explored (Wang & Ba, 2020) for policy distillation from CEM, but only largely sub-optimal policies could be extracted. When the policy is used alone at test time and not in combination with the MPC-CEM optimizer, its performance drops

significantly, becoming almost random for some environments. We argue that the reason behind the difficulty in distilling a policy from CEM expert data is the multimodality of the CEM solution space and its inherent stochasticity due to the sampling, which we address below.

Another possibility to train higher-performance policies from experts is DAgger (Ross & Bagnell, 2010), which is an on-policy method, asking the expert to relabel/correct policy actions. Nevertheless, as we will show in the following sections, DAgger alone is not sufficient to extract high-performing policies in our setting. To solve this problem, we use a guiding cost in combination with DAgger. A combination of GPS and DAgger-style relabeling was proposed in PLATO (Kahn et al., 2017), however to create unbiased training data from iLQG experts. Since unguided DAgger is not appropriate with CEM, PLATO is not successful in our setting either. The components of our algorithm will be explained in the following sections.

## 3 METHODS

Trajectory optimization aims to find a suitable action sequence $\vec{a}_t = (a_t, a_{t+1}, \dots, a_{t+h})$ of horizon $h$ that minimizes a cost function $f(\vec{a}_t, s_t)$, where $s_t$ is the current state of the system. i.e.

$$\vec{a}_t^\star \leftarrow \arg\min_{\vec{a}_t} f(\vec{a}_t, s_t). \tag{1}$$

The cost function $f$ encodes the task. Optimal control is obtained if $f$ is the trajectory cost until step $h$ plus the cost-to-go under the optimal policy for an infinite-horizon problem, evaluated at the last state in the finite horizon trajectory. Typically, the cost-to-go is replaced by a proxy cost such as the distance to a target. In our case, the cost $f$ is given by the sum of negative environment rewards up to the planning horizon.

### 3.1 IMPROVED CROSS-ENTROPY METHOD FOR MPC

To optimize Eq. 1, we use a variant of the Cross-Entropy Method. Although originally introduced as an adaptive importance sampling procedure to estimate rare-event probabilities, it was recently employed in model-based RL (Chua et al., 2018; Wang & Ba, 2020) as a trajectory optimizer.

A practical implementation of CEM uses a Gaussian proposal distribution over the optimization variables $\vec{a}$, $\mathcal{N}(\mu, \sigma)$, evaluates the cost function $f(\vec{a})$, and refits this distribution iteratively by using the top-$k$ samples. It finds low-cost regions of $f$ and high-performing samples of the variable $\vec{a}$. The cost function can be either expressed through a learned value function or through the dynamics model of the system, which in turn can be learned from data or in an analytical form. In the model-predictive control (MPC) framework, CEM can be used at every time-step to generate the optimal action plan, of which only the first action is executed, and the whole procedure is repeated for the next steps until task completion.

In this paper, we will make use of iCEM, a sample-efficient improvement of CEM by Pinneri et al. (2020) that makes use of *colored-noise* and *memory*. In particular, it generates correlated action sequences with a non-flat frequency spectrum, differently from the flat Gaussian noise of CEM. This, together with the memory addition, results in an order of magnitude sample reduction.

#### 3.1.1 USING A POLICY TO INFORM THE OPTIMIZATION: iCEM$_\pi$

Unguided search for action sequences can be hard, in particular, if the cost function has large flat regions (sparse rewards). Since we aim at extracting a policy $\pi(s)$, we can expect that after some examples the policy can be used to guide the search into the right region. Thus, we warm-start the mean $\mu$ of the iCEM Gaussian distribution with the policy actions. In particular, at time $t = 0$ where no prior information exists, the mean is initialized from rolling out the trajectory with the policy until the planning horizon: $\mu \leftarrow (\pi(s_0), \pi(s_1), \dots \pi(s_h))$. Whenever the action for a new step is computed, iCEM uses shift initialization of the mean, and only the mean-action at the end of the horizon is initialized from the policy. Since iCEM is sample-based we also provide it with samples from the policy directly. More concretely, the actions performed by the policy are added in the last iteration if iCEM. The policy-informed iCEM algorithm is called iCEM$_\pi$ and is shown in Alg. 1.

---

**Algorithm 1:** Improved Cross-Entropy Method with warm-starting and samples from policy $\pi$, denoted as iCEM$_\pi$. Blue marks policy *warm-starting* and red is *adding policy samples*.

---

**Input:** $f(\vec{a}, s)$: cost function; $\pi$: policy; $N$: # of samples; $h$: planning horizon; $k$: elite-set size;
$\quad\quad\quad$ $\beta$: colored-noise exponent; $\sigma_{\text{init}}$: noise strength; *iter*: # of iterations; $\gamma$: reduction factor.

1   $\tau \leftarrow \emptyset$
2   **for** $t = 0$ **to** $T-1$ **do**
3     $s \leftarrow$ get the current state from the environment
4     **if** $t == 0$ **then**
5       $\mu_0 \leftarrow h$-steps rollout of model with $\pi$ starting from $s$
6     **else**
7       $\mu_t \leftarrow$ shifted $\mu_{t-1}$ with last time-step action given by $\pi$
8     $\sigma_t \leftarrow$ constant vector in $\mathbb{R}^{d \times h}$ with values $\sigma_{\text{init}}$
9     **for** $i = 0$ **to** *iter*$-1$ **do**
10       $N_i \leftarrow \max(N \cdot \gamma^{-i}, 2 \cdot k)$
11       samples $\leftarrow N_i$ samples from clip$(\mu_t + \mathcal{C}^\beta(d, h) \odot \sigma_t^2)$ ;      // with $\mathcal{C}^\beta(d, h)$ colored-noise
         Normal distribution with noise-exponent $\beta$ and dimension $(d, h)$
12       add a fraction of the shifted/reused elite-set to samples
13       **if** $i == $ *last-iter* **then**
14         add $\mu_i$ to samples
15         add policy actions to samples ;      // $h$-step rollout with policy from current state $s$
16       costs $\leftarrow$ cost function $f(\vec{a}, s)$ for $\vec{a}$ in samples
17       elite-set$_t \leftarrow$ best $k$ samples according to costs
18       $\mu_t, \sigma_t \leftarrow$ fit Gaussian distribution to elite-set$_t$ with momentum
19     $a \leftarrow$ first action of best elite sequence
20     add $(s, a)$ to $\tau$
21     execute $a$
22   **return** $\tau$;                                                // return the trajectory

---

## 3.2   OFF- AND ON-POLICY IMITATION LEARNING

How difficult is it to clone a policy from iCEM$_\pi$? Using a basic off-policy method like Behavioral Cloning (BC), which simply does point-to-point L2 loss minimization, does not work out-of-the-box. This is because during test-time, the policy visits state-space regions for which it never received any expert feedback, and it consequently fails. This problem is also known as "covariate shift". The performance gap (difference in cost to go) between the expert and the policy scales quadratically in time due to the covariate shift (Ross & Bagnell, 2010).

A classical way to address this problem is with on-policy imitation learning. DAgger (Dataset Aggregation) by Ross et al. (2011) is the most popular example. DAgger works by iteratively rolling out the current policy, querying the expert for the correct actions (also called relabeling) on the states visited by the policy, and training the policy on the aggregate dataset across all iterations. The covariate shift is alleviated by populating the data with states visited by previous iterations of the policy.

**ISSUES OF STOCHASTIC TEACHERS:** Nevertheless, some problems arise when considering population-based optimizers as experts, both for BC alone or in combination with DAgger. The optimized action sequence of CEM might converge to different solutions every time it is asked to relabel the policy actions. Naturally, many actions can result in an equally high-performing solution. As an example: in the Fetch Pick & Place task, well before grasping the box, the action of opening or closing the gripper can be arbitrary, as long as the action of opening the gripper is executed on time. To illustrate this problem, we analyze the variance of the relabeled actions produced by the same expert on a fixed state sequence. The variance is high, spanning over a third of the action-space, as can be seen in Fig. 2(a).

## 3.3   GUIDED POLICY SEARCH

To lift some of the aforementioned problems, the expert can be guided by the current policy. For high-dimensional control tasks, in addition to the high variance (Fig. 2(a)), zero-order optimizers may

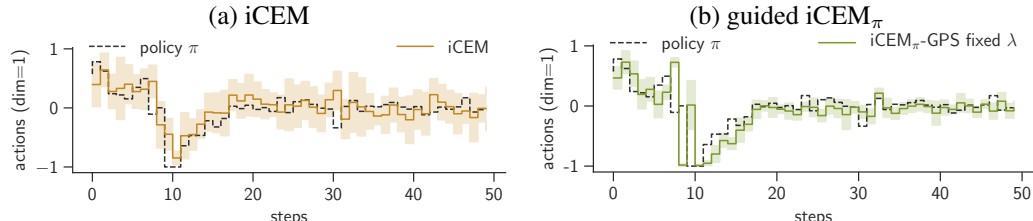

Figure 2: Variance of DAgger actions when relabeling 10 times the same trajectory in case of unguided iCEM (a) vs. guided iCEM$_\pi$ (b), for the FETCH PICK&PLACE task. The action variance of iCEM is considerably higher than the one of iCEM$_\pi$-GPS guided by the shown policy (std-dev=0.30 vs 0.13). The policy $\pi$ is trained from a single expert rollout.

also find multiple different solutions, leading to multi-modal training data. This prohibits successful policy extraction through supervised learning. To address these issues, the cost function $f$ in Eq. 1 becomes the main cost $J$ (task) plus a guidance cost to bias the optimizer's solution towards the policy:

$$\vec{a}_t^\star \leftarrow \arg\min_{\vec{a}_t} J_t(\vec{a}_t, s_t) + \lambda D_{\mathrm{KL}}(\pi_\theta || \vec{a}_t), \tag{2}$$

which is also known as Guided Policy Search (GPS) (Levine & Koltun, 2013). When competing solutions arise, the one closer to the policy is preferred, tackling in this way the problem of multi-modality and creating much more consistent training data. As a remark, in our experiments, we use deterministic policies which reduces the KL-divergence to a squared difference between the actions.

In addition to Eq. 2, we further guide iCEM by warm-starting it from the policy, as implemented in iCEM$_\pi$ (Alg. 1). As a consequence, both together lead to a reduced action variance, as seen in Fig. 2(b). For an illustrative state sequence, the standard deviation of expert actions drops from $\sigma = 0.30$ without guidance to $\sigma = 0.13$.

The data for training the policy comes from the expert interacting with the environment, that we call *expert rollouts*. When combining GPS with DAgger we use the same cost-function (Eq. 2) for the normal expert rollout and for asking the expert to perform action relabeling. This contrasts with PLATO (Kahn et al., 2017), designed for iLQG experts, where the guidance cost is only used for the expert rollouts. Also, in PLATO the policy is not executed to collect new data, but relabeling of guided expert data is done using the unmodified cost to collect optimal training data. For zero-order optimizers, unguided solutions are not helpful as shown above. Without relabeling the actual policy rollouts, we found PLATO not to work in our setting.

The hyperparameter $\lambda$ in Eq. 2 is difficult to choose. It might be adapted according to learning progress or other heuristics. However, when using the guided cost in CEM other considerations have to be made which lead to a simple adaptation scheme introduced in the next section.

### 3.4 ADAPTIVE AUXILIARY COST WEIGHTING

The purpose of $\lambda$ is to trade-off the potential loss in cost by staying close to the policy. Generalizing Eq. 2, we can phrase the cost function $f$ in Eq. 1 as the sum of the main cost $J$ and a set of auxiliary costs $C_j^{\mathrm{aux}}$:

$$\vec{a}_t^\star \leftarrow \arg\min_{\vec{a}_t} J_t(\vec{a}_t, s_t) + \sum_j \lambda_j C_j^{\mathrm{aux}}(\vec{a}_t, s_t) \tag{3}$$

Examples of auxiliary costs are the guidance cost as in Eq. 2 and the action norm $\|\vec{a}\|$ to prefer small action magnitudes. We follow the philosophy that the auxiliary costs are subordinate and should only bias the solutions without causing a large performance drop. This leads to the idea that the auxiliary costs should never dominate the optimization. Since the main costs might vary by orders of magnitude, for instance, in sparse reward settings, we opt for a formulation where the auxiliary costs can maximally lead to a fixed fractional loss in performance.

As a motivating example, let us consider hard tasks that present flat regions in the cost function: in this case, a non-zero $\lambda$ can lead to a failure of guided CEM to find any good solution. How is that possible? Let us consider the Fetch Pick & Place task as shown in Fig. 1. When none of the sampled

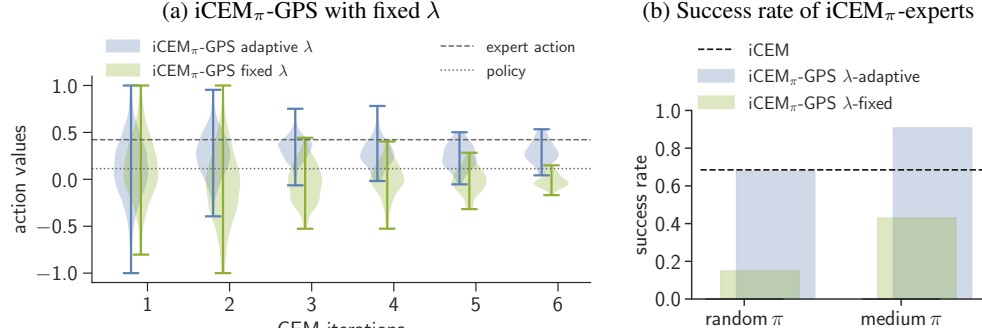

Figure 3: Effect of adaptive $\lambda$ throughout iCEM$_\pi$ iterations and success rate on the FETCH PICK&PLACE task. **(a)** The action sampling distribution is shown over the iCEM-iterations (at a predefined time-step) and one of the 4 action-dimensions when guiding with a weak policy. Dashed lines indicate the action of the policy and of a high compute-budget iCEM expert. Fixed $\lambda$ shifts the distribution too early, resulting in a collapse to the policy behavior and failure to find a good solution. **(b)** Average success rate of iCEM$_\pi$ expert (low compute-budget with 45 samples) over 800 episodes.

action-sequences moves the box (which is quite likely in the first CEM iteration) then all sequences are equivalent under the main cost. Elites (Alg. 1 line 17) are only selected based on the auxiliary costs resulting in a quick reduction in sampling variance and preventing a good solution to be ever found. To illustrate this phenomenon, Fig. 3 displays the sampling distribution of iCEM$_\pi$ throughout the optimization iterations (line 9 in Alg. 1). Using a fixed $\lambda$ (iCEM$_\pi$-fixed $\lambda$) typically converges to the policy action which is often unsuccessful in a new situation or when the policy is still weak.

We propose the following adaptation of $\lambda_j$:

$$\lambda_j = c_j \frac{\mathcal{R}(J)}{\mathcal{R}(C_j^{\text{aux}}) + \epsilon} \tag{4}$$

where $c_j$ is the new hyperparameter, $0 < \epsilon$ is a small regularization to avoid amplifying tiny auxiliary costs, and $\mathcal{R}(C)$ represents the cost-range of $C$ in the elite set:

$$\mathcal{R}(C) = \max_{\text{elite-set}} C - \min_{\text{elite-set}} C . \tag{5}$$

This formulation ensures that $\lambda_i$ is zero if the main cost $J$ cannot be improved by the preliminary elite-set (as in the example above). When an actual improvement is possible and the cost landscape is not flat ($R(J) > 0$), the auxiliary cost influences up to a fixed fraction ($c_j$) of the cost-range. Thus, the sampling distribution for the adaptive case will converge only if a good solution is found, as visualized in Fig. 3 (a) (iCEM$_\pi$-GPS adaptive $\lambda$) for one auxiliary cost being the KL-term as in Eq. 2. The new hyperparameter $c$ is easier to tune than $\lambda$. We use the same $c$ for all experiments, although the main costs are 3 orders of magnitude apart.

Another aspect of the ratio in Eq. 4 is that, when none of the elite sequences is close to the policy network, then the denominator of Eq. 4 is small, which heavily steers the sampling towards the policy, however, *only* if the numerator is not zero ($R(J) \neq 0$), i.e. a reward signal from the environment is detected. Figure 9 in appendix D shows how the adaptive $\lambda$ parameter evolves over time in the FETCH PICK&PLACE environment for a weak and medium policy.

**TEACHER IMPROVEMENT:** Another effect of adaptive guidance in combination with iCEM$_\pi$ is the improvement of the expert/teacher, which is not common nor expected in standard imitation learning. The increased success-rate is shown in Fig. 3(b). As we can see from the plot, the guidance alone is not sufficient to reach the original iCEM expert performance: when the auxiliary cost becomes adaptive, the iCEM benefits from the policy.

### 3.5 PUTTING THE PIECES TOGETHER: APEX

Our method, named Adaptive Policy EXtraction (APEX), uses adaptive policy-guided iCEM$_\pi$ and DAgger using the same expert to create data for successful training of policies in an iterative fashion.

---

**Algorithm 2:** Adaptive Policy EXtraction procedure (APEX)

---

**Input:** iCEM$_\pi$;    $\pi_\theta$: policy network; $n$: # rollouts per iteration
1  init $\theta$ randomly;
2  $\mathcal{D} \leftarrow \emptyset$;
3  $i \leftarrow 1$
4  **while** *not converged* **do**
5      $f(\vec{a}, s) \leftarrow J_t(\vec{a}, s_t) + \lambda_1 D_{KL}(\pi_\theta \| \vec{a}) + \lambda_2 \|\vec{a}\|$;            // see Eqns. 2, 3, and 4
6      $\tau_{CEM} \leftarrow n$ Rollout with iCEM$_\pi(f(\vec{a}, s), \pi_\theta)$;          // $\tau$ is the resulting trajectory
7      add $\tau_{CEM}$ to $\mathcal{D}$
8      $\theta \leftarrow$ train policy $\pi_\theta$ on $\mathcal{D}$
9      $\tau_\pi \leftarrow$ Rollout with $\pi_\theta$
10     $\tau_{DAgger} \leftarrow$ relabel actions in $\tau_\pi$ with iCEM$_\pi(f(\vec{a}, s), \pi_\theta)$ ;      // DAgger
11     add $\tau_{DAgger}$ to $\mathcal{D}$
12     $\theta \leftarrow$ train policy $\pi_\theta$ on $\mathcal{D}$
13     $i \leftarrow i + 1$ ;                           // one APEX iteration
14 **return** $\pi_\theta$

---

The main steps are: create data from the guided expert, update the policy, rollout the policy and relabel the actions using the same expert (DAgger), add the relabeled data to the dataset, and update the policy. This iterates until the desired performance is reached. More details are found in the pseudo-code in Alg. 2.

## 4   RESULTS

Can strong policies be obtained from model-based planning and imitation learning? By considering high-dimensional challenging robotics environments, some of them with sparse rewards, we test APEX and find that the answer is yes. In some cases, we considerably improve the state-of-the-art currently held by model-free methods.

We perform our experiments on a selection of 4 environments, listed below, which use the MuJoCo (Todorov et al., 2012) physics engine:

**HUMANOID STANDUP:** (OpenAI Gym Brockman et al. (2016) v2) A humanoid robot is initialized in a laying position. The goal is to stand-up without falling, i.e. reaching as high as possible with the head. We use a task horizon of 500.

**FETCH PICK&PLACE (sparse reward):** (OpenAI Gym v1) A robotic manipulator has to move a box, randomly placed on a table, to a randomly selected target location. The reward is only the negative Euclidean distance between box and target location, so without moving the box there is no reward.

**DOOR (sparse reward):** (DAPG project Rajeswaran et al. (2018)) A simulated 24 degrees of freedom ShadowHand has to open a door with a handle. The reward is the sum of door opening, a quadratic penalty on the velocities, and a bounty for opening the door (in contrast to Rajeswaran et al. (2018) where a guidance of the hand to the handle was used).

**HALFCHEETAH RUNNING:** (OpenAI Gym v3) A half-cheetah agent should maximize its velocity in the positive x-direction. In contrast to the standard setting, we prohibit a rolling motion of the cheetah by heavily penalizing large absolute angles of the root joint. In the standard setting, numeric instabilities in the simulator are exploited by iCEM.

In Fig. 4, we compare APEX against several imitation learning baselines: Behavioral Cloning (BC) (iCEM BC), DAgger (iCEM DAgger), and BC from iCEM$_\pi$ with guidance cost (fixed $\lambda$) and warm-starting (iCEM$_\pi$-GPS). For reference, we also provide the performance of SAC[1] (Haarnoja et al., 2018) as a model-free RL baseline, to get an idea of the difficulty of the learning tasks. Note that we use partially sparse rewards settings in FETCH PICK&PLACE (only box to goal reward) and DOOR (sparse reward) (no hand-to-handle reward) which makes them particularly challenging. However, for FETCH PICK&PLACE, a specialized method for goal-reaching tasks using hindsight relabeling (DDPG+HER) (Andrychowicz et al., 2017) can solve the task. In HUMANOID STANDUP

---

[1] We took the implementation from https://github.com/vitchyr/rlkit

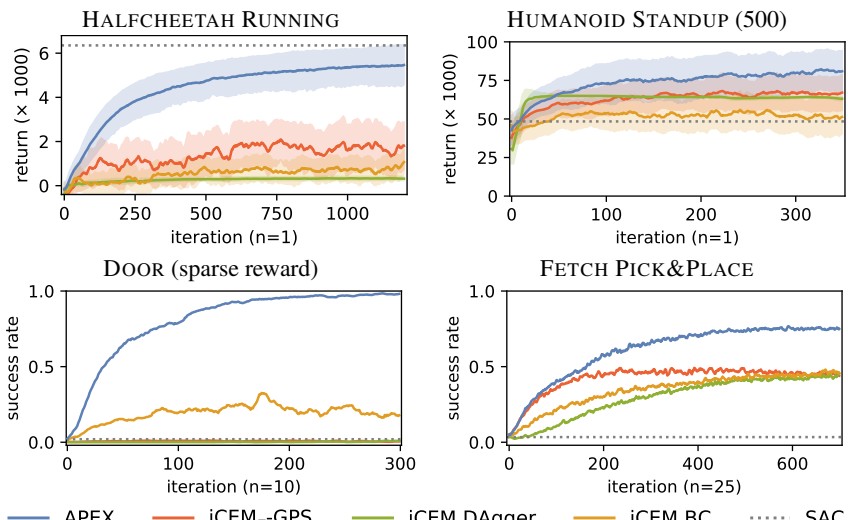

Figure 4: Policy performance on the test environments for APEX and baselines. SAC performance is provided for reference.

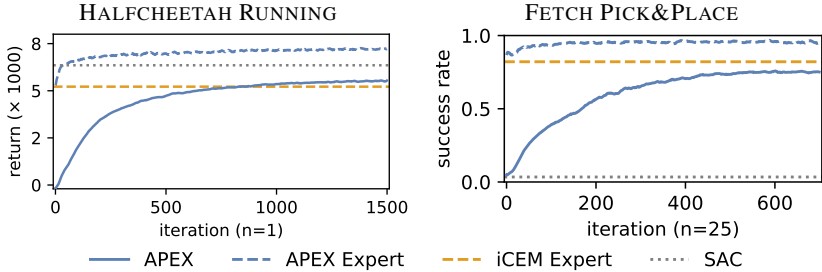

Figure 5: Interplay between policy and expert. Policy performance (solid line) and expert performances (dashed lines) on selected test environments for APEX. Due to warm-starting and adding policy samples, experts improve with the policy. For low budgets this effect is stronger, see Fig. 8.

the performance of SAC marks the behavior of just sitting. This is a local optima and is hard to escape for gradient-based methods. APEX manages to stand up but cannot balance for long, presumably because there are many ways the robot can fall. Notably, the DOOR (sparse reward) environment with its 24 DoF Shadow hand, is solved by our method with a high success-rate.

A very interesting effect of our approach is that the iCEM$_\pi$ expert working inside of APEX improves with the policy, as seen in Fig. 5. When guided by a policy, the performance raises (dashed blue lines) with the policy performance, even if the policies themselves are not very strong yet. Together, APEX is able to shrink the gap between expert and policy performance and yields strong results, see also the illustrative behaviors in Fig. 1. The policy can achieve significantly higher performance than iCEM, as prominent in the low-budget case in HALFCHEETAH RUNNING presented in Fig. 8.

## 4.1 ABLATIONS

Are all components of APEX required? We perform several ablations to investigate this question. Figure 6 shows the performance when removing each individual component from APEX, as detailed in Appendix B. Using the adaptive guidance cost instead of a *fixed* $\lambda$ is only important in the sparse reward environments, as expected. In case of dense rewards, the $\lambda$ adaptation does perform identically. Removing warm-starting has a drastic effect in all environments. APEX without DAgger is also much worse. Not adding policy samples to the optimization has an interesting effect. At first glance the policy performance is higher, however, asymptotically the full APEX is better or on par. The reason is shown in Fig. 7 where we report the expert performance of APEX and APEX without policy

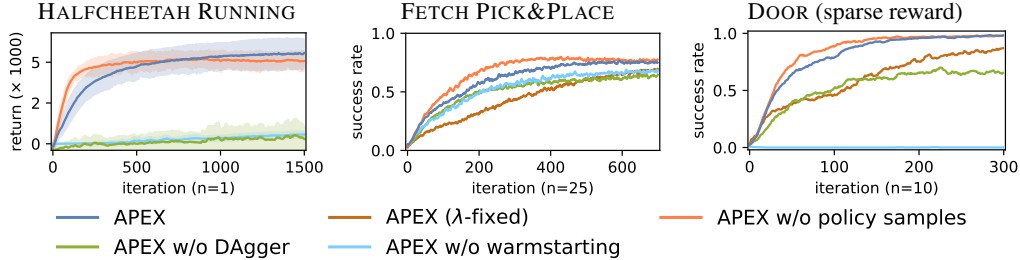

Figure 6: Ablation experiments. We remove different components of the APEX algorithm, see legend. In case of HALFCHEETAH RUNNING, the performance for APEX with $\lambda$-fixed is not reported as it matches that of APEX.

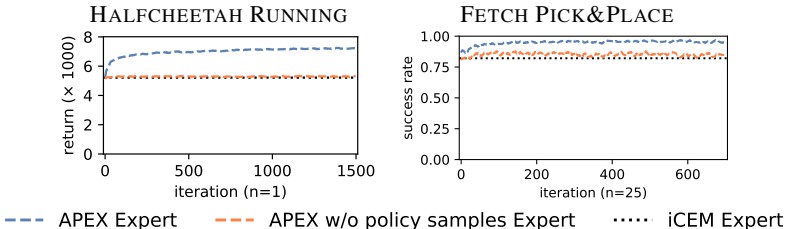

Figure 7: The expert performance for APEX and APEX without adding policy sample. As seen, the expert performance improves with learnt policy as the added policy sample directs expert distribution towards a better solution space.

samples. The expert in the latter case only marginally improves upon the standard iCEM. Thus, the policy is limited to the iCEM performance, whereas in APEX it can go beyond, see Fig. 5.

## 5 CONCLUSIONS

We study the problem of policy extraction from model-based trajectory optimizers using CEM – a zero-order blackbox method – popular in recent model-based RL. Our method (APEX) is able to extract strong policies for hard robotic tasks which are especially challenging for model-free RL methods. This is achieved by imitation learning from a guided CEM expert, where both policy and expert mutually propel themselves to higher performances.

With this work, we want to propose a promising stepping stone towards learning high-performing policies for real robots, where speed and minimal interaction with the system are more important than asymptotic performance. Model-based RL methods are very sample efficient (Chua et al., 2018; Wang & Ba, 2020) in terms of actual environment interactions, but are too slow to run in real-time yet. With the recently proposed improved CEM (Pinneri et al., 2020), the possibility for real-time planning became feasible but still needs huge computational resources. Extracting high performing policies is a promising route to success, where we provide here an important ingredient. Although we are using the simulator as an internal model to study the policy extraction problem in isolation, there are good chances that our method will transfer to the model-based RL setup with learned models which we will investigate in future work. One reason for being optimistic about the transfer is that the core problems with stochastic optimizers are identical. Another reason is that a powerful optimizer like CEM ends up finding brittle solutions that are overfitted to the dynamics of the simulator and make imitation hard. Learned models can potentially prevent this from happening.

Another route to real robot applications is to pretrain models and policies in simulation and then adapt to the real system and extract policies with APEX. Similarly, the sim-to-real transfer can be done without learned models but using the ground-truth simulations as done here to get a strong initialization for fine-tuning on the real hardware. One might ask: why adopting a model-based planning method if we are using only a simulator? Because planning with population-based optimizers can produce nearly-optimal solutions and the extracted policy (if able to match expert's performance) can beat model-free baselines like SAC, which can get stuck in local minima in high-dimensional systems, as happened in the complicated HUMANOID STANDUP task.

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

APPENDIX

## A    PERFORMANCE TABLES

We report the performance numbers for our experiments in Tab. 1, while the expert/teacher settings are indicated in Tab. 2, and policy settings are in Tab. 3. Wall-clock times are in Tab. 4.

Table 1: Performances for all environments for APEX and APEX policy. Enviroments are abbreviated for space reasons: HC-Run: HALFCHEETAH RUNNING, Hum-Up: HUMANOID STANDUP, and FPP: FETCH PICK&PLACE. We report the cumulative reward (marked with [1]) and the success rate (marked with [2]). SAC and iCEM baseline performances are provided for reference. For an explanation about the budget, see Pinneri et al. (2020).

| Envs | APEX (budget 45) | | (budget 100) | | (budget 300) | | SAC |
|---|---|---|---|---|---|---|---|
| | Expert | Policy | Expert | Policy | Expert | Policy | |
| HC-Run[1] | 5556±241 | 4847±665 | 7202±510 | 5596±779 | 9310±437 | 6218±1337 | 6352 |
| Hum-Up[1] | 149.0k±9.6k | 73.5k±11.9k | 173.7k±13.4k | 80.5k±14.1k | 207.3k±4.5k | 88.7k±12.3k | 48.4k |
| FPP[2] | 0.893 | 0.0311 | 0.947 | 0.770 | – | – | 0.034 |
| DOOR[2] | 1.0 | 0.99 | – | – | – | – | 0.02 |

| Envs | iCEM (budget 45) | (budget 100) | (budget 300) |
|---|---|---|---|
| HC-Run[1] | 3488±119 | 5235.9±167 | 7632.54±250 |
| Hum-Up[1] | 146.3k±13.7k | 182.2k±12.5k | 202.4k±51.6k |
| FPP[2] | 0.67 | 0.81 | – |
| DOOR[2] | 1.0 | – | – |

Table 2: Expert settings for the considered methods (The values for colored noise exponent for different environments are taken from Pinneri et al. (2020))

| | # elites $K$ | Initial std. $\sigma_{init}$ | Momentum $\alpha$ | Decay $\gamma$ | Fraction reused elites $\zeta$ | Guidance scaling constant $c$ | Horizon $h$ |
|---|---|---|---|---|---|---|---|
| iCEM | 10 | 0.5 | 0.1 | 1.25 | 0.3 | – | 30 |
| iCEM$_\pi$ | 10 | 0.5 | 0.1 | 1.25 | 0.3 | – | 30 |
| APEX | 10 | 0.5 | 0.1 | 1.25 | 0.3 | 0.025 | 30 |

| | Warm Start | Add Policy Sample | # rollouts per iteration $n$ | | |
|---|---|---|---|---|---|
| iCEM | False | False | – | | |
| iCEM$_\pi$/ APEX | True | True | 1 (HALFCHEETAH RUNNING, HUMANOID STANDUP) 10 (DOOR), 25 (FETCH PICK&PLACE) | | |

Table 3: Policy settings for iCEM$_\pi$ and APEX

| # layers | Size | Activation fn | l1 reg. | l2 reg. | Optimizer | Learning rate |
|---|---|---|---|---|---|---|
| 3 | 128 | ReLu | 1e-6 | 1e-5 | Adam | 5e-4 |

| Batch size | Iterations | # latest rollouts used for training | | | | |
|---|---|---|---|---|---|---|
| 1024 | 1000 | 50 (HALFCHEETAH RUNNING), 100 (HUMANOID STANDUP) 150 · 25 (FETCH PICK&PLACE), 100 · 10 (DOOR) | | | | |

Table 4: Wall-clock times for APEX per iteration. Reference machine for wall clock-times is an Intel Xeon Gold 6154 CPU @ 3.00GHz using 32 Cores for parallel simulation models. All times are in minutes.

| Env | Budget | Expert-only | Dagger-only | Training | Total |
|---|---|---|---|---|---|
| | | Wall-clock times per iteration (in min.) | | | |
| HALFCHEETAH RUNNING | 45 | 1.8 | 2.0 | 1.4 | 5.6 |
| | 100 | 2.3 | 2.5 | 1.4 | 6.5 |
| | 300 | 4.0 | 4.2 | 1.5 | 10.1 |
| HUMANOID STANDUP | 45 | 1.7 | 2.5 | 0.9 | 5.3 |
| | 300 | 5.6 | 6.9 | 0.9 | 13.8 |
| FETCH PICK&PLACE (25 rollouts) | 45 | 7.78 | 8.7 | 3.2 | 20.0 |
| | 100 | 12.2 | 13.5 | 3.5 | 29.5 |
| DOOR (10 rollouts) | 45 | 9.7 | 11.2 | 3.9 | 25.1 |

## B  ABLATION EXPERIMENTS

In order to understand which components of APEX are affecting the performance of the policy extraction and the expert, several ablations were carried out. The results of this ablation studies are shown in Fig. 6 and Fig. 7. In the following, the implementation details of the different ablation studies are discussed.

APEX ($\lambda$-fixed): Instead of using the adaptive scheme for $\lambda_j$ that were proposed in Eq. 4, its value is set to a constant value, with a value chosen to work well in the respective environment.

APEX w/o DAgger: Instead of using DAgger for policy extracting, plain behavioral cloning is used. In DAgger, states visited by the policy are relabeled with actions from the expert and added in addition to the expert data to the training dataset of the policy. In our ablation, only the data from the expert is added to the training dataset of the policy. In case of DAgger, the policy gets twice the amount of data as in the case of behavioral cloning because in each iteration data from the expert and relabeled data from the policy is added to the dataset. In both cases, the same number of gradient-steps are performed during training of the policy.

APEX w/o warmstarting: Instead of initializing the means of all action dimensions along the planning horizon with the actions from the policy at the beginning of each rollout, means along action dimensions and the planning horizon are initialized with zero. After each planning step, the mean of action $t$ is not initialized with the action from the policy but repeats the last action at time-step $t-1$.

APEX w/o policy samples: No additional sample trajectory with actions from the policy is added to the other samples during planning (see Alg. 1). To clarify in the full version with policy samples one of the random samples is overwritten to keep $N$ samples.

## C  EXPERT AND POLICY INTERPLAY

In Fig. 8 we report the performance of the policy and the experts, for different compute budgets. As discussed in the main paper, it is interesting that the expert inside APEX improves with the policy. The can lead to a higher policy performance than the original iCEM with the same compute budget.

## D  ADAPTIVE $\lambda$ IN GPS COST

In Fig. 9, we report how the adaptive $\lambda$ parameter in the GPS cost of APEX changes during planning for a weak and medium policy, respectively, in case of FETCH PICK&PLACE. In both runs, we fixed the goal and target locations. In case of the week policy, no successful solution is found in the first few time steps, meaning the min cost is the same as the max cost among elites. Hence, $\lambda$ is zero allowing the optimizer to freely explore without being hampered by the weak policy. In case of the

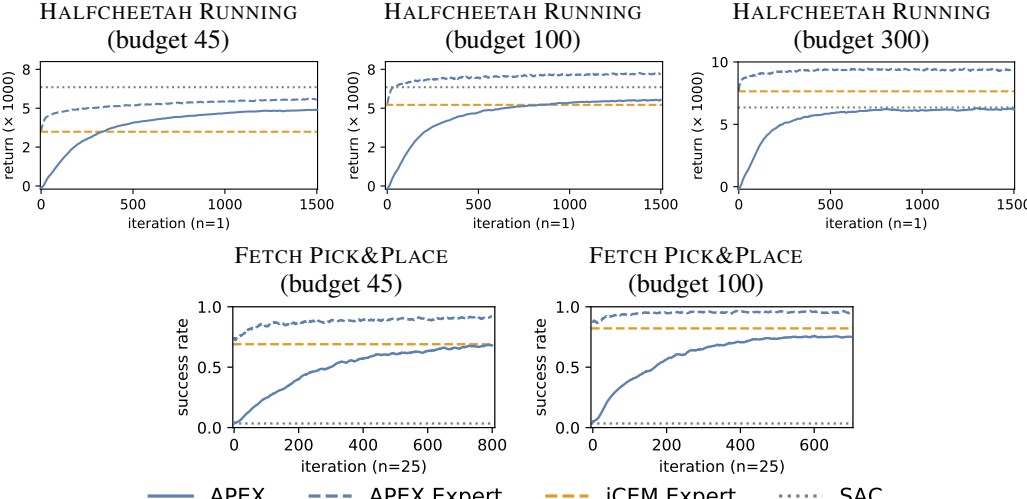

Figure 8: Same as Fig. 5 but for different compute-budgets: 45, 100 (normal), 300. Notice, that in the case of low and normal budgets in HALFCHEETAH RUNNING the policy outperforms the iCEM expert. In FETCH PICK&PLACE the policies are able to match the iCEM performance.

better policy, a solution is found early on, possibly because of the optimizer being guided by the policy. Thus $\lambda$ becomes non-zero in the second step already.

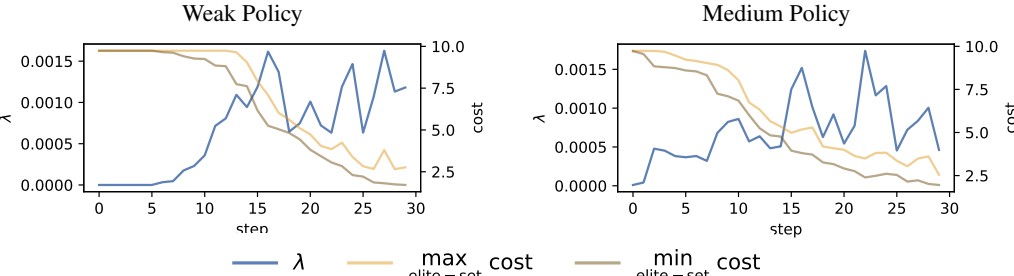

Figure 9: Evolution of the adaptive $\lambda$ parameter during planning. Left for a weak, right for a medium policy. The light and dark orange curves show the original min/max cost ($J$) among the elites. The blue curves show how lambda changed due to the differences in the original costs.

