# OpenReview forum: "Extracting Strong Policies for Robotics Tasks from Zero-Order Trajectory Optimizers"
_ICLR.cc/2021/Conference — ICLR 2021 Poster_

### Official Review · AnonReviewer1 · 2020-10-24
**review1**

**Rating:** 6
**Confidence:** 3

**Review:**

This paper presents a method to combine zero order optimizer with imitation learning. Several components are essential for good performance, including policy guided initialization, DAgger and adaptive auxiliary loss.

Overall the paper provides a clear description of the essential components of the algorithms and the result is also quite strong.

Several comments below:

(1)The right most figures in Figure 4 is really hard to unpack with so many information and almost similar colors. Honestly I don't understand these figures, it would be nice if the authors make it clearer, either in the caption or make cleaner figures.

(2) For the first figure in Figure 5, the text said " it performs
initially better than with adding the samples, but the experts do not improve with the policy in this
case such that the asymptotic performance is lower". But the figure shows it has the same final performance as Apex, am I missing something here? Also more distinguished color coding can make the figures easier to digest.

(3) Maybe related to the previous question, what does the learning curve shows? Is it performance of the policy or the performance of the ICEM warm starting with the policy?

(4) How important is an accurate dynamics? All the experiments are done with perfect dynamics information, which I also believe is deterministic, which is not possible in real world scenario. Some experiments with dynamics noise added or with learned model in the CEM rollouts can make the paper's claim stronger.

(5) half cheetah is not a difficult task as claimed in the introduction of the paper. The paper itself shoes SAC can already perform very well. Actually this task is a very strange choice compared to other tasks considered. I would recommend something that is more like the others.

---

> ### Author Response · Authors · 2020-11-21
> **Answer to Reviewer 1**
>
> Thank you for your review and your positive evaluation of our work. We have used the comments to improve the paper. See also our general answer above.
>
> 1) We agree that the plots were difficult to understand. We unpacked them now into 2 figures and deferred some detail to the appendix. Please check the updated paper.
>
> 2) We cleared up what we meant with that sentence by adding Fig.7. It is shown that the expert in APEX is only noticeably improving upon iCEM if policy samples are added. That is why the APEX policy can improve upon "APEX without adding policy samples". We are currently running longer training to show this effect clearer. As additional evidence, in Fig 8 (Appendix) you can see how for a low budget the APEX policy improves significantly above the iCEM-expert performance.
>
> 3) All solid curves show pure policy performance. Dashed lines show expert performances (with or without policy warm-starting etc). We made that clearer in the caption.
>
> 4) Thank you for the suggestion. We could not run extensive experiments with noisy dynamics now, but preliminary results show that it plays more in our favor. We added a discussion into the conclusion section on this aspect.
> A perfect model can also play against policy extraction because the iCEM expert is extremely good at exploiting the simulator and finding highly specialized and overfitted solutions. We can provide two primary examples:
>
> - in the HalfCheetah environment, the optimizer (even plain CEM) finds an unphysical rolling/flipping motion by exploiting the coarse time discretization of the system
> (that eventually leads to ever-increasing speeds until the simulation crashes when using the optimizers). This flipping motion is very hard to learn by a policy via imitation learning. It is also found by a model-free optimizer like SAC but not by model-based baselines like PETS-CEM. We think that the rationale is that CEM couldn't converge to that kind of solution when the model is inaccurate and takes into account uncertainty.
>
> - in the Fetch Pick and Place task: Sometimes the task is solved not by fetching, picking, and placing the box to the goal, but by snapping it to the ground with one finger and let it fly to the goal. Of course, that can only happen if the agent has perfect information about the dynamics. Trying to learn a policy from such a demonstration is extremely hard and we argue that an imperfect (or rather, stochastic) model can help generate more robust solutions.
>
> The multimodal and stochastic nature of the optimizer will not go away with learned models so we believe our method will transfer well. However, as there are also challenges with learning good models, isolating problems seems reasonable to us.
>
> 5) We agree, the HalfCheetah is not a hard robotic task but it was one of the environments to fail in the POPLIN paper, where they tried for the first time to extract a policy from CEM. We thought it was important to report the results also on this task. We changed the wording.

---

### Official Review · AnonReviewer4 · 2020-10-28
**An interesting paper that combines GPS and DAGGER for imitating CEM's policy**

**Rating:** 5
**Confidence:** 3

**Review:**

In summary, this paper combines GPS and DAgger to learn a policy network by imitating a model-based controller, which uses iCEM as the optimizer. Their approach uses both the iCEM controller and the learned policy with DAgger-like relabeling to collect data and then train the network with behavior cloning. An auxiliary loss inspired GPS, which encourages the consistency between the expert controller and the learned policy, together with an adaptive weighting method, is added to reduce the multi-modal issue. The experiment results show promising results.

Strong points:
1. Distilling/imitating CEM policy seems to be a promising approach for solving some control problems, especially when the model is known, or we can learn a good model. Given the deterministic model, CEM can find solutions much faster than an RL algorithm, using CEM to search demonstrations and distilling them into a policy network sounds like a very promising approach in the future.
2. The authors discuss the issue of stochastic teachers and the multi-modal training data and solve them by combing GPS and DAgger. Their approach is simple yet effective.
3. The adaptive auxiliary cost weighting is novel to me.

Weak points:
1. The whole pipeline is not very novel to me. The paper simply adds GPS into DAgger by replacing the LGQ with CEM, which is straightforward if one wants to combine optimal control with neural networks.
2. However, unlike GPS or other model-based DRL papers, this paper requires the ground truth model, which largely limits its application. I agree it's still important to find a way to first solve the problem in the simulation, but I hope the author can give some words about this assumption.
3. I don't think that the sparse reward and the flat regions matter in the model-based RL setting, where the cost is usually dense and designed by the user. Given the limited search horizon, for example, in PickAndPlace, if the agent can only receive rewards when the object is close to the goal, and the planning horizon is short, it's unlikely for a CEM controller to find a solution. Why not simply changing the reward to avoid those flat regions?
4. This work studies how to do imitation learning when the expert comes from a stochastic teacher. However, no imitation learning baseline except the simplest BC is considered. Approaches like GAIL are not evaluated. I don't think  Wang & Ba (2020) can prove that GAN is not suitable in this case. As far as I know, they use the learned model instead of the ground truth model, which is different from the setting in this paper. The authors should at least compare with some STOA imitation learning methods.

Based on the previous weakness, like the loss of the novelty, the missing imitation learning baseline, and the limited applications, I wouldn't recommend acceptance given the current version.

Minor issues:
1. Section 3. "Optimal control is obtained if f is the trajectory cost until step h plus the cost-to-go under the optimal policy for an infinite-horizon problem, evaluated at the last state in the finite horizon trajectory. " I believe some words are missing here. Do you have a value network like POLO in your optimization?
2. Ablation study. APEX without DAgger? What's that? Does this mean no relabeling in Algorithm 2? What are the differences between that and the one without policy samples? How can you train the policy by sampling the trajectories with the policy network while not relabeling it?
3. What's the time complexity? In Table 1. of the appendix, some results are blank. Does this mean the computation complexity is still too high to finish them in time?
4. What do the iterations mean for SAC? Is this comparison fair? I think with the same number of environment episodes, SAC has much fewer environment steps. CEM needs extra environment steps to search for a solution.
5. Presentation is not very clear. For example, what does "expert rollout" means in section 3.3? It's hard for me to guess its meaning without reading the Alg 2. in sec 3.5. There are also some grammar mistakes.

Possible ways to improve:
1. Maybe the authors can combine their approaches with a learned model like PETS-CEM. It would be great if their approach could also benefit in the model-based RL setting.
2. Provide more imitation learning comparisons.

---

> ### Author Response · Authors · 2020-11-21
> **Answer to Reviewer 4**
>
> Thank you very much for your helpful review that we used to improve our paper.
> See also our general answer above.
>
> Answer to "weak points":
>
> 1) When we tried the first time to extract a policy from iCEM we realized that the simple combination of methods was not enough. Namely, using GPS and DAgger, like done in PLATO, was producing very poor performing policies. The reasons are stochasticity and multimodality of CEM. GPS partially fixes the first problem, but not the second.
> With this paper, we primarily want to shed some light on why it is *not* straightforward and eventually propose a pipeline that works, even where model-free algorithms fail.
>
> 2) Yes, as you said perfect dynamics was necessary for an unbiased analysis. But, as we already said to Reviewer 1 (answer 4) and Reviewer 3 (answer 3), having a ground truth model is *not necessary* for APEX to work. We only used them to have fewer moving parts. We have seen that having a perfect model and a nearly-optimal optimizer can produce very performing but also very brittle solutions, which are much harder to learn by a policy. The problems we identified with policy extraction are identical in the learned-model case. We updated the Conclusions.
>
> 3) The sparse/semi-sparse reward setting is the largely preferred way for a user to actually specify the task. Dense reward versions are hand-engineered guesses on how good guidance looks like. In many cases, they cause unwanted side-effects and have to be tuned carefully.
> Thus the sparse rewards are important, also in the model-based case, and remain a big challenge. Sampling-based optimizers are a good choice when the user does not want to design a cost by hand. We used semi-sparse rewards (e.g. object to goal distance, but not hand to object cost) in the experiments to make a comparison with SAC, and to show that model-based planning can rival model-free methods.
> However, we agree that providing suitable proxy costs can only help iCEM and APEX.
>
> 4) In POPLIN [1] they show different approaches to policy extraction. In some cases, GAN training performs much better than BC. We actually do not aim at comparing with imitation learning (IL) baselines: our method can be applied to whatever IL framework.
> But as you proposed, it could have been interesting to show that APEX can potentially benefit from other forms of IL like GAN training. Nevertheless, this was outside the scope of our contribution and we encourage other researchers to use APEX inside of their pipelines.
>
> Answer to minor issues:
>
> 1) We have no value function. We clarified the statement.
>
> 2) It means that lines 10 and 11 of Alg. 2 (APEX) are not executed. The policy is trained only with off-policy behavioral cloning, while policy samples are still used by the expert. We explain the ablations in detail in the newly added Appendix B.
>
> 3) The higher the budget the larger the performance, however, it is most interesting to get strong performance with relatively cheap (low sample number) iCEM. Currently, the computational budget is dominated by ground-truth simulations. Their speed depends on the system complexity. As shown in the iCEM paper, with learned models the speed can be real-time for the expert. We report wall-clock times in the appendix now.
>
> 4) Reported SAC performance is at convergence, so it is just used for reference, as stated in the caption. The iteration number is not referring to SAC, that is why it is a horizontal dotted line.
>
> 5) Thank you for bringing this up. We introduce this in a better way now.
>
> [1] Exploring model-based planning with policy networks, Tingwu Wang \& Jimmy Ba, ICRL 2020

---

> > ### Comment · AnonReviewer4 · 2020-11-24
> > **Response**
> >
> > Thanks for your response. The paper looks better now, and I agree the paper has solid contributions. However, I don't think "believing" the approach could be combined with a learned model/other IL methods is enough. I would suggest you mention them as future works instead of conclusions.

---

> > > ### Author Response · Authors · 2020-11-24
> > > **Response to Reponse**
> > >
> > > Thank you for your response and for checking our paper again.
> > > You are right, we have changed to formulation to phrase it as future work and give reasons why this can work.

---

### Official Review · AnonReviewer2 · 2020-10-28
**Distillation of a planner expert into a policy; More analysis and baselines needed.**

**Rating:** 6
**Confidence:** 3

**Review:**

This paper presents an approach to distill a model-based planning expert into a policy to enable real-time execution on robotic systems. An improved version of the Cross-Entropy Method, iCEM is used to generate trajectories via model-based optimisation where the forward model is the ground-truth simulator dynamics. The expert is further improved by warm-starting using the learned policy. The policy is learned via imitation learning of trajectories from the expert. Several approaches for this step are presented, including the vanilla approach of behaviour cloning (BC), dataset aggregation (DAGGER) to reduce covariate shift and Guided Policy Search (GPS) where the planner is encouraged to keep close to the policy distribution via a trust region KL loss. The paper discusses the merits of each approach and proposes Adaptive Policy Extraction (APEX), an approach that learns the policy from the expert through a combination of DAGGER and GPS where the tradeoff between the cost terms on planner exploration and policy trust region KL is set adaptively. This approach is tested on four continuous control tasks and achieves good performance compared to baselines — the learned policy gets significant improvements compared to the baselines. Additionally, a study that ablates different parts of the APEX algorithm is also presented.

Pros:
1. This paper tackles the problem of distilling model-based planner trajectories into a policy. This is a key problem in model-based RL where planner + policy proposals can often exceed the performance of the policy but the former can be too computationally expensive to run on the real system. The proposed approach takes some steps towards this by combining ideas from prior work (GPS, DAGGER, iCEM) to improve policy learning on challenging continuous control tasks.
2. The paper is well written and motivated. The experiments are well structured and provide insights into the struggles of BC and other imitation learning methods when combined with stochastic optimisers like CEM. Different ideas such as DAGGER and GPS are introduced and the combination of these together with adaptive weighting is clearly motivated.
3. The proposed approach leads to significant improvements compared to baseline model-free methods and other imitation learning variants on the presented tasks. Additional experiments including ablation studies also establish further intuition on different parts of the proposed method.

Cons:
1. The proposed approach primarily combines ideas from prior work including GPS and DAGGER. One piece of novelty is the adaptive setting of the \lambda parameter — this is set to be a fixed scalar times the ratio between the range of the main task cost and the range of the auxiliary costs. When all trajectories get the same or similar main task reward (as can happen in sparse reward tasks) this goes to zero, thereby avoiding collapsing the planner to the policy and leading to premature convergence. As evidenced in the experiments, it is not clear if this adaptation helps anywhere else (and it also comes across as a bit hacky). If this is the case, why not use a simpler adaptation scheme that just sets the lambda to zero when the task reward is uniform and a fixed lambda otherwise? In general though, I find the adaptive lambda not well motivated. It would be useful if there is more analysis on the behaviour of this adaptation throughout learning — maybe a plot of \lambda throughout learning can shed more light.
2. The plots in Fig 4 and Fig 5 are quite hard to parse — there are a few places where the colors of the lines are flipped (see eg. Door in both figures, the APEX line is miscoloured). Additionally, there are a few lines that are missing/appear extra in some plots (see eg. Bottom middle plot in Fig 4 where the dashed line is not relevant & bottom right where some lines are missing). It would be great if these can be fixed.
3. Looking at the results in Fig 4 it looks like the expert (iCEM + Policy proposal) is still ~2x better than the learned policy even with the proposed approach. What is the reason for this? Can this gap be shortened further? It is a bit unsatisfying that the policy cannot match the performance of the planner.
4. There are no strong model-based baselines provided apart from different variants of imitation learning coupled with iCEM (which are essentially ablations of the proposed method APEX). It would be great if a prior method which shows results on the Cheetah (Wang et al, 2020 from the paper) can be compared with the proposed approach. This should give a better understanding of the relative strengths of the proposed method compared to the state of the art.
5. As the paper mentions, one key limitation of the approach is the use of a GT dynamics model. A next step could be to extend the approach to train models jointly with the policy. It would be useful to have a longer discussion on this in the paper.

Overall, the paper presents some interesting ideas and the results are promising. More analysis on the adaptation scheme and better baselines are needed to significantly strengthen the paper. I would suggest a borderline accept.

---

> ### Author Response · Authors · 2020-11-21
> **Answer to Reviewer 2**
>
> Thank you for the thorough review and that you like our paper. Please also check our general answer above.
> Regarding your comments listed under Cons:
>
> 1) We see our novelty also in the integration, namely that the optimizer can improve with the policy, that policy learning is stable, etc. Regarding the adaptation of lambda: it helps to avoid premature convergence, as you correctly remarked, which is supported by several experiments (Fig. 3, 4, and 6).
> We believe that avoiding policy collapse is an important improvement over the baselines.
>
> Why our formulation and were does it help?
> In addition to the effect above, our formulation allows specifying a more intuitive hyperparameter: the relative impact of auxiliary costs. If the main cost changes during the task over one order of magnitude (very common in sparse reward settings) a fixed lambda value is hard to define. We updated the text to motivate this better.
>
> As regards the lambda evolution, we find it a great idea so we added a plot showing how it changes during a typical rollout at the beginning and later in learning, see Appendix D.
>
> 2) Thank you for pointing that out, we changed them. Now Fig. 4 is split into two and some details are in the appendix. Fig 5 uses a more visible color scheme now.
>
> 3) At the time of submission our runs were not completed, we are sorry for that. We updated it (now Fig 5 and Fig 8) which now shows that the policy can match the performance of the original iCEM expert and improve upon it in some cases. In cases where we cannot catch up, we think it is mostly due to iCEM finding brittle behaviors that are hard to imitate.
>
> 4) and 5)  The reason which motivated this paper was to understand why it is so difficult to extract a policy from a stochastic optimizer like CEM. This was also shown in POPLIN [1], where only for simple environments, training the policy with Behavioral Cloning was successful. Their Cheetah performance was already not able to keep up with the expert’s one (below 500 for BC training - around 1600 for GAN training). For more complicated environments their policy performance “is almost random”. Note that for those same environments the expert reached very high performance (with learned model), but the policy still lagged behind by several orders of magnitude. Since we decided to focus only on matching the expert’s performance, we did this in isolation to the model learning problem.
>
> [1] Exploring model-based planning with policy networks, Tingwu Wang \& Jimmy Ba, ICRL 2020

---

### Official Review · AnonReviewer3 · 2020-11-01

**Rating:** 6
**Confidence:** 2

**Review:**

The paper focusses on how to extract good policies from experts in imitation learning scenarios. The paper proposes a series of trick for policy extraction from Cross-Entropy-Method-based (CEM/iCEM) trajectory optimizers. The algorithm integrates iCEM with DAgger and Guided Policy Search.
The extracted policies show good performance versus a model-free baseline (Soft-Actor Critic) and various other extraction baselines (Behavior Cloning, DAgger, Guided Policy Search).

Strengths:
1. The motivation of the paper is clear. In particular, a pretty thorough analysis of the state of the art and its limitations motivates the current approach
2. Experiments are conducted on various standard benchmarks (Humanoid Standup, Half Cheetah, DOOR, Fetch Pick& Place) including sparse reward domains
3. Results are convincing
4. An ablation study shows how different parts lof the proposed method do interact with different environments


Weakness/Comments:
1. It seems that this is primarily a combination of existing methods. What exactly is the new contribution apart from recombination.
2. Why does not adding policy samples have a counter-intuitive effect in HALFCHEETAH RUNNING?
3. Why do you think this will transfer to model-based RL with learned models

---

> ### Author Response · Authors · 2020-11-21
> **Answer to Reviewer 3**
>
> Thank you for your comments that we used to improve our paper. Please also check the general answer above. We would like to clarify your doubts:
>
> 1) Apart from combining GPS and DAgger (similar to PLATO) our contribution is to integrate it with zero-order optimizers like CEM (iCEM): This required to address the problem of stochastic multimodal optimizer, something that might be of interest in a wider context.
> In addition, our integration allows the optimizers to improve through the policy, also a novelty.
> We agree that this was not sufficiently explained in the paper, despite being very important, so we added an explanation at the end Sec. 3.4 about the teacher improvement to better address your concerns.
>
> 2) We cannot explain precisely why APEX without policy samples is initially performing better. One reason might be that initially policy actions are "winning" in the iCEM optimization and are not providing new learning signals.
> Importantly, however, the expert is not improving over iCEM significantly if policy samples are not added. We added figure 7 to show this effect. The updated plots better show asymptotic behavior.
>
> 3) In general, our purpose was to extract strong policies from the trajectories produced by a population-based optimizer, rather than assessing the policy’s performance on learned models. In other words, to close the gap between the performances of experts and policy. In POPLIN, in fact, the learned policy performance is still, in the best case, one order of magnitude lower than CEM.
> Nevertheless, with our new insights and our method, we believe that our method will transfer. Learned models can actually be beneficial for a simple reason: CEM will not be able to exploit the simulator, producing overfitted and brittle solutions. As a result, the expert CEM actions will be easier to imitate by a policy.
>
> We added a more detailed discussion about learned-models and how our method can be used in the conclusion section.

---

### Author Response · Authors · 2020-11-21
**General Answer**

We thank the reviewers for their positive reviews and the constructive criticism which we used to improve and refine the paper. In particular:

- we made the plots easier to parse, both visually and conceptually, respectively with additional figures and further explanations (added Fig. 5, better color code in Fig. 6, added Fig. 7, added, added Fig. 8 and section C, added Fig. 9 and section D)

- we updated the rest of the figures with longer training curves (some runs are not yet finished and will be updated of course)

- we clarified the problem of transferring to learned models by adding a paragraph in the conclusions, thanks to reviewer 1, 3 and 4

- we added a section in the appendix on the adaptive lambda and its evolution in time, which was an interesting suggestion by reviewer 2

- we added a section in the appendix explaining the ablations in details

---

### Decision · Program_Chairs · 2021-01-07
**Final Decision**

**Decision:**

Accept (Poster)

**Comment:**

This paper proposes a method to solve high-dimensional, continuous robotic tasks offering a trajectory optimization and a distill policy. The paper is well-written and the work is promising. It is very relevant for the robotics and RL communities.